# Atmospheric Nitrogen Dioxide Improves Photosynthesis in Mulberry Leaves via Effective Utilization of Excess Absorbed Light Energy

**Yue Wang, Weiwei Jin, Yanhui Che, Dan Huang, Jiechen Wang, Meichun Zhao and Guangyu Sun *** 

School of Life Sciences, Northeast Forestry University, Harbin 150040, Heilongjiang, China; wangyue@nefu.edu.cn (Y.W.); jinwei6677066@163.com (W.J.); carcar@nefu.edu.cn (Y.C.); 18804501876@163.com (D.H.); 15771398049@163.com (J.W.); 15846070751@163.com (M.Z.)
* Correspondence: sungy@nefu.edu.cn; Tel.: +86-0451-8219-1507

**Abstract:** Nitrogen dioxide ($NO_2$) is recognized as a toxic gaseous air pollutant. However, atmospheric $NO_2$ can be absorbed by plant leaves and subsequently participate in plant nitrogen metabolism. The metabolism of atmospheric $NO_2$ utilizes and consumes the light energy that leaves absorb. As such, it remains unclear whether the consumption of photosynthetic energy through nitrogen metabolism can decrease the photosynthetic capacity of plant leaves or not. In this study, we fumigated mulberry (*Morus alba* L.) plants with 4 $\mu L \cdot L^{-1}$ $NO_2$ and analyzed the distribution of light energy absorbed by plants in $NO_2$ metabolism using gas exchange and chlorophyll a fluorescence technology, as well as biochemical methods. $NO_2$ fumigation enhanced the nitrogen metabolism of mulberry leaves, improved the photorespiration rate, and consumed excess light energy to protect the photosynthetic apparatus. Additionally, the excess light energy absorbed by the photosystem II reaction center in leaves of mulberry was dissipated in the form of heat dissipation. Thus, light energy was absorbed more efficiently in photosynthetic carbon assimilation in mulberry plants fumigated with 4 $\mu L \cdot L^{-1}$ $NO_2$, which in turn increased the photosynthetic efficiency of mulberry leaves.

**Keywords:** nitrogen dioxide; nitrogen metabolism; photorespiration; heat dissipation; excess absorbed light energy; electron transfer; photochemical efficiency

## 1. Introduction

Atmospheric nitrogen oxides ($NO_x$) mainly include nitric oxide (NO), nitrogen dioxide ($NO_2$), dinitrogen trioxide ($N_2O_3$), dinitrogen monoxide ($N_2O$), and dinitrogen pentoxide ($N_2O_5$). Nitrogen oxides other than $NO_2$ are extremely unstable and can be converted to $NO_2$ in the presence of light, humidity, or heat [1]. $NO_2$ sources are divided into natural and man-made sources. Natural sources mainly include lightning, stratospheric photochemistry, and microbiological processes in ecosystems. $NO_2$ formed in nature is generally in ecological balance at a natural point of equilibrium, which is low relative to man-made air pollution [2]. The emission of $NO_2$ from man-made sources is indeed the main component of atmospheric pollutants, which forms aerosol particles of nitric acid with particulate matter in the air. These aerosols form secondary pollution with pollutants from sources that include fossil fuel and biomass combustion as well as various electroplating, carving, welding, and other industrial emissions [3,4].

$NO_2$ not only causes acid rain, but also changes the competition and species composition among wetland and terrestrial plant taxa, reduces atmospheric visibility, increases acidification and eutrophication of surface water, and increases the toxin content of fish and other aquatic organisms [5]. Cheng et al [6] found that tiny particles of water in the air act as incubators during hazy conditions and trap $NO_2$ to interact and form sulfate. Additionally, stationary polluted weather systems accelerate

chemical reactions, trapping near-surface $NO_2$, leading to $NO_2$ concentrations that are more than three times higher than that found in sunny weather. This increase in aerosol mass concentration leads to an increase in water content, accelerating the accumulation of sulfate and causing severe haze. In addition, $NO_2$ is also a respiratory system irritant. After being inhaled, $NO_2$ first affects the respiratory organs, the lungs in particular. The combination of nitrite and nitric acid that occurs when $NO_2$ encounters mucus membranes is a strong irritant with corrosive effects [7].

$NO_2$ affects the normal growth of plants. When $NO_2$ concentrations are higher than the annual average $NO_2$ concentration limit of 53 ppb in the United States [8], $NO_2$ can damage the leaves of plants, causing chlorosis in angiosperms, needle burns in conifers [9,10], reduced leaf area [11], and lower stem weights [12]. However, when the concentration of $NO_2$ is lower than the average annual $NO_2$ concentration of 53 ppb in the United States, the total leaf area, nutrient intake, and aboveground biomass were more than doubled [13,14]. Similar results have been found in different plant species, including *Arabidopsis thaliana* [15,16], tobacco (*Nicotiana plumbaginifolia* L.) [17], and crops such as lettuce (*Lactuca sativa* L.), sunflower (*Helianthus annuus* L.), cucumber (*Cucumis sativus* L.), and squash (*Cucurbita moschata* L.) [9]. In addition, atmospheric $NO_2$ can shorten flowering periods in tomato (*Solanum lycopersicum* L. 'Micro-Tom'), increasing the number of flowers and the yield of the fruit [18].

In China, the concentration of $NO_2$ emission, which caused formation of fine particulate matter (PM2.5), is not enough to injure tree plants. As a result, trees have been used to absorb atmospheric nitrogen dioxide to reduce atmospheric PM2.5 [19].

Photosynthesis is the foundation of plant growth and development, and the primary source of photosynthetic energy is light [20]. When the absorption of light energy is excessive, the excess excitation energy can harm the photosynthetic systems, causing photosynthetic inhibition, and even photooxidation and photodamage. Plants have multiple photoprotective mechanisms that reduce the potential harm of excess light energy to the photosynthetic apparatus under strong light. Reducing excess energy in addition to heat dissipation dissipates energy by other means [21]. Nitrogen metabolism and photorespiration also use and consume light energy or photosynthetic electrons absorbed by leaves [22,23]. However, it is unclear whether this consumption of photosynthetic energy will reduce the photosynthetic capacity of plant leaves and thereby hinder the growth and development of plants.

Mulberry, which has strong adaptability to the environment, has been an important economic tree species in China since ancient times. Nowadays, mulberry can be used for sericulture, new high-protein forage grass, fruit tree, and greening trees in northeast China [24–26]. We have studied the response of mulberry to atmospheric pollutant $SO_2$ and found that mulberry is very sensitive to it [24]. We also studied the response characteristics of mulberry to nitrogen and obtained significant results in the absorption and metabolism of nitrogen by mulberry trees [27–30]. We think atmospheric $NO_2$ also affects the nitrogen metabolism of mulberry trees. In the present study, we fumigated mulberry leaves with $NO_2$ and assessed the impact on nitrogen metabolism, photorespiration, photosynthetic energy distribution, and electron flow distributions. Our assays characterized the response of photosynthetic efficiency to light energy used and consumed by mulberry plants, including the absorption of $NO_2$ via nitrogen metabolism in vivo.

## 2. Materials and Methods

### 2.1. Plant Material and Growth Conditions

Seedlings of *Morus alba* L. were selected as experimental materials. Mulberry seeds were provided by the Heilongjiang Sericulture Institute of Heilongjiang Province in China. Seeds with strong, full, and uniform size were selected, disinfected with 75% ethanol for 3min, rinsed with distilled water for 5–6 times, and soaked with distilled water at 25 °C for 24 h. The seeds were blotted with sterile filler paper and sown into the seedling tray. Two seeds were sown in each hole. After germination and cultivation, the test seedlings were grown to a height of 10 cm and then transplanted into pots with a diameter of 12 cm and a height of 15 cm. Experimental seedlings were cultured in a seedling greenhouse with an average temperature of 28/25 °C (light/dark), light intensity of 400 µmol m$^{-2}$·s$^{-1}$,

photoperiod of 12 h/12 h (light/dark), and relative humidity of about 75%. The culture substrate was uniformly mixed with peat and vermiculite, and irrigated with 800 mL of tap water every 2 days. In order to ensure relative consistency of the experimental materials, the branches and leaves of the mulberry seedlings were removed at the time of transplantation, such that only 5 cm of the main root and the main stem were preserved. Two plants were planted into each pot, with 80 plants in total cultivated. When the seedlings had grown to a height of 30–40 cm, 12 mulberry seedlings with uniform growth were selected for $NO_2$ fumigation treatment.

## 2.2. $NO_2$ Fumigation Treatment

The gas fumigation box was a custom-made open-top automatic monitoring gas concentration device [31]. The $NO_2$ cylinder (Dalian Date Gas, Dalian, China) was connected to an electromagnetic valve and axial flow fan as the air supply source. Solenoid valve controls regulated the test gas, while the axial flow of the fan is used to send the test gas into the air chamber. A gas pressure reducer and a gas flowmeter in the middle of the gas delivery system control the gas flow. A solenoid valve associated with the micro-flux switch valve controls the average flow in low-frequency pulse width modulation (PWM) mode. The gas chamber was composed of organic glass in the shape of a six-sided prism, with a cross section diagonal length of 1.16 m and height of 1.85 m. Vent holes were set on the top, with a gas grid plate 0.30 m from the bottom of the chamber. The $NO_2$ gas first enters the bottom space under the gas grid, and then passes through the gas chamber from the bottom of the 1200 holes, each with a diameter of 2 mm and evenly distributed across the grid plate. A fan was installed at the top of the fume chamber to ensure that the conditions were evenly distributed among the plants. The concentration of $NO_2$ used in the experiment was 4 $\mu L \cdot L^{-1}$, and the concentration was selected according to our pre-experiment results. We used 0.5, 1, 2, 4, 6, 8 $\mu L \cdot L^{-1}$ $NO_2$ to fumigate mulberry leaves. The results showed that the net photosynthetic rate in leaves of mulberry was the highest as $NO_2$ concentration 4 $\mu L \cdot L^{-1}$. The fumigation time is 4 hours/day, from 7:30 to 11:30; the temperature in the fumigation chamber was maintained at 28 °C. Samples were taken before the fumigation began (0 h) and at 4 and 8 h after fumigation. The indicators of N metabolism and photosynthetic gas exchange parameters as well as chlorophyll *a* fluorescence parameters were determined.

## 2.3. Measurement of $NO_3^-$-N Content

An analysis of nitrate nitrogen was completed by using the salicylic acid method [32]. Standard solutions of 500 mg/L nitrate nitrogen with deionized water to create a 20, 40, 60, 80, and 100 mg/L series of standard solutions. Then, 0.5 g of plant material was placed into each test tube, to which 10 mL of deionized water was added before being placed into a boiling water bath for 30 min. Afterwards, test tubes were cooled with tap water, and extracts were filtered into volumetric flasks. The residues were rinsed, and the samples were allowed to finally settle to 25 mL. Then, 0.1 mL aliquots of the above series of standard solutions and sample extracts were respectively transferred into new test tubes, and the standard solution was replaced with 0.1 mL of distilled water.

Then, 0.4 mL of 5% salicylic acid solution was added prior to each sample being shaken well with a 20 min of room temperature incubation. After incubation, 9.5 mL of 8% NaOH solution was added, and samples were shaken and allowed to cool to room temperature. The total volume of the sample was 10 mL. The absorbance was measured at 410 nm with a blank as a reference. A standard curve was drawn, and a regression equation was fitted to the standard by using the nitrate concentration as the abscissa and the absorbance as the ordinate. The concentration of nitrate nitrogen was calculated using the inferred regression equation, and content was calculated by using the formula $NO_3^-$-N content $= (C \times V/1000)/W$, where *C* is the regression equation calculated $NO_3^-$-N concentration, *V* is the extract total mL of sample liquid, and *W* is the sample fresh weight.

## 2.4. Measurement of Amino Acid Content

A colorimetric analysis was done on hydrated ninhydrin for amino acids [33]. To assess amino acid content, 200 µg/mL amino acid standard solutions were measured into volumes of 0.0, 0.5, 1.0, 1.5, 2.0, 2.5, and 3.0 mL, respectively, and transferred into 25 mL volumetric flasks, to which water was added until a volume of 4.0 mL was reached. Then, 1 mL of ninhydrin solution (20 g/L) and 1 mL of phosphate buffer (pH 8.04) was added to each flask, followed by mixing and incubation in a 90 °C water bath until the color became constant. At this point, samples were removed rapidly and cooled to room temperature, and water was added until a final volume was reached. Samples were then shaken well and allowed to stand for 15 min. The absorbance A of the remaining solutions was then determined relative to a reagent blank as a reference solution at a wavelength of 570 nm. In the standard curve, the micrograms of amino acids represented the abscissa, while absorbance A was the ordinate. The standard curve was drawn based on these data, and the regression equation was then inferred.

Then, 0.5 g plant samples were added to 5 mL of 10% acetic acid and ground in a mortar. These ground samples were then washed into 100 mL volumetric flasks, diluted to a set volume in water, and filtered into triangle bottles for determination of the filtrate. Then, 4 mL of the clarified sample solution was subjected to the standard curve inference procedure outlined above, in which the absorbance A value was measured under the same conditions and the microgram content of amino acids was calculated using a regression equation of the following form: amino acid content (µg/100 g) = $C/(m \times 1000) \times 100$, where $C$ represents the mass number of amino acids and the $m$ represented the mass of the sample.

## 2.5. Measurement of Nitrate Reductase Activity

Nitrate reductase (NR) activity was determined using the kit produced by Suzhou Keming Company (Suzhou, China). This kit functions by assaying how NR catalyzes the reduction of nitrate to nitrite according to the reaction $NO_3^- + NADH + H^+ \rightarrow NO_2^- + NAD^+ + H_2O$. The resulting nitrite quantitatively produces red azo compounds under acidic conditions with p-aminobenzenesulfonic acid and $\alpha$-naphthylamine. The generated red azo compound has a maximum absorption peak at 540 nm and was measured by spectrophotometry.

## 2.6. Measurement of Nitrite Reductase Activity

Nitrite reductase (NiR) activity was determined using the kit produced by Suzhou Keming Company. This kit functions by assaying how nitrite reductase reduces $NO_2^-$ to NO, and the sample is subjected to the diazotization reaction to generate a purple-red compound that indicates a $NO_2^-$ decrease; that is, a change in absorbance at 540 nm reflects the activity of nitrite reductase.

## 2.7. Measurement of Gas Exchange Parameters

Gas exchange was measured with a portable photosynthesis system (LICOR-6400; LI-COR, Lincoln, NE, USA). All the photosynthetic measurements were taken at a constant airflow rate of 400 µmol·s$^{-1}$, and the temperature was $28 \pm 2$ °C. While using the liquefied $CO_2$ cylinders were used to provide different $CO_2$ concentrations, across a $CO_2$ concentration gradient of 50, 100, 200, 300, 400, 600, 800, 1000, and 1200 µmol·mol$^{-1}$. The photosynthetically active radiation (PAR) was 1000 µmol m$^{-2}$·s$^{-1}$ in order to avoid light limitation of photosynthesis. The photosynthetic rate ($P_n$) and stomatal conductance of $H_2O$ ($G_s$) were measured. Photosynthesis curves plotted against intercellular $CO_2$ concentrations ($P_n$-Ci curve) was analyzed to estimate the maximum carboxylation rate ($V_{cmax}$) of ribulose-1,5-bisphosphate carboxylase/oxygenase (Rubisco) and dark respiration rate ($R_d$) [34].

## 2.8. Measurement of Chlorophyll A Fluorescence Transient and Light Absorbance at 820 nm

According to the method developed by Schansker et al. [35], leaves were first dark-adapted for 20 min, and then the rapid chlorophyll fluorescence-induced kinetic curve (OJIP curve) and 820 nm light absorption curve were measured using a plant efficiency analyzer (M-PEA, Hansatech,

King's Lynn, UK). The OJIP curve was induced under 3500 μmol·m$^{-2}$·s$^{-1}$ pulsed light, and the fluorescence signal was recorded from 10 μs to the end of 2 s, with an initial recording rate of $10^5$ data per second. The relative value of the difference between the maximum ($I_o$) and minimum ($I_m$) absorbance at 820 nm, i.e., $\Delta I/I_o = (I_o - I_m)/I_o$, was used as an index for measuring photosystem I (PSI) activity. The OJIP fluorescence induction curve was analyzed with reference to the JIP-test used by Strasser et al. [36] to measure the initial fluorescence ($F_o$), the maximum fluorescence ($F_m$), the light energy absorbed by the unit reaction center ($ABS/RC$), the unit reaction center captured for the reducing energy of $Q_A$ ($TR_o/RC$), energy captured by the unit reaction center for electron transfer ($ET_o/RC$), and energy dissipated in the unit reaction center ($DI_o/RC$). The potential activity of photosystem II (PSII) was then calculated as $F_v/F_o = (F_m/F_o) - 1$. The maximum photochemical efficiency of PSII was calculated as $F_v/F_m = 1 - (F_o/F_m)$. Lastly, the number of active reaction centers per unit area was calculated as $RC/CS_m = (F_v/F_m)(V_J/M_o)(ABS/CS_m)$.

## 2.9. Measurement of Photochemical Quenching, Electron Transfer Rate, and Absorbed Energy of the PSII Reaction Center

After dark adaptation of the leaves for 30 min, chlorophyll fluorescence quenching analysis was performed using a pulse modulation fluorimeter (FMS-2, Hansatech, King's Lynn, UK). In this procedure, the leaf adapts to the light for the first 30 s, and the light intensity is the same as the ambient light intensity before the measurement of the leaf (1000 μmol·m$^{-2}$·s$^{-1}$). The steady-state fluorescence parameter ($F_s$) and electron transfer rate (ETR) under the light-adapted condition was measured, and the saturated pulsed light (8000 μmol·m$^{-2}$·s$^{-1}$) was used to determine the photochemical quenching (qP), the maximum fluorescence value ($F_m'$). Then, the method of Hendrickson et al. [37] was followed to determine the light energy absorbed by the PSII reaction center into each of four parts [38]. That is, the following parameters were inferred: quantum yield ($\Phi_{PSII}$) for the photochemical reaction, quantum yield ($\Phi_{NPQ}$) dependent on the proton gradient lutein cycle on both sides of the thylakoid membrane, quantum yield of fluorescence and heat energy dissipation ($\Phi_{f,D}$), and the thermal dissipation of the quantum yield ($\Phi_{NF}$) of the deactivated PSII reaction center. These parameters were inferred using the following formulae:

$$\Phi_{PSII} = [1 - (F_s/F_m')][(F_v/F_m)/(F_v/F_{mM})]$$
$$\Phi_{NPQ} = [(F_s/F_m') - (F_s/F_m)][(F_v/F_m)/(F_v/F_{mM})]$$
$$\Phi_{f,D} = (F_s/F_m)[(F_v/F_m)/(F_v/F_{mM})]$$
$$\Phi_{NF} = 1 - [(F_v/F_m)/(F_v/F_{mM})].$$

The sum of each of these parameters is 1, namely $\Phi_{PSII} + \Phi_{NPQ} + \Phi_{f,D} + \Phi_{NF} = 1$ (Figure 1).

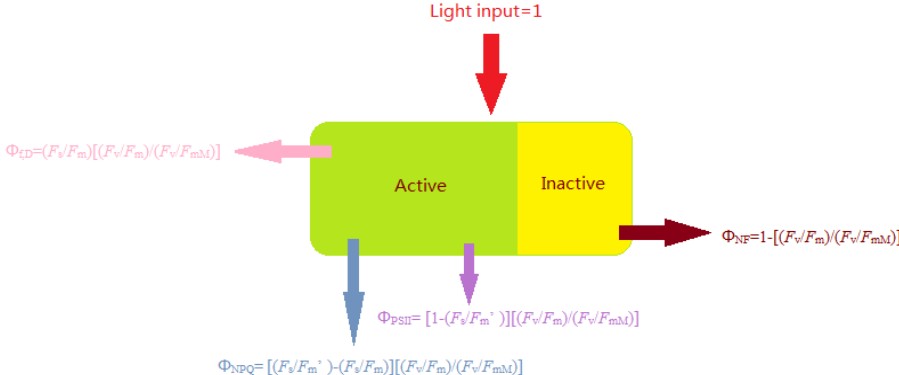

**Figure 1.** Energy allocation pathways of photosystem II (PSII): proportion of light energy used in the quantum yield of PSII photochemistry ($\Phi_{PSII}$), xanthophyll-mediated thermal dissipation ($\Phi_{NPQ}$), quantum yield used in thermal dissipation in non-functional PSII ($\Phi_{NF}$), and quantum yield of fluorescent and heat energy dissipation ($\Phi_{f,D}$).

### 2.10. Statistical Analysis

Excel and SPSS software were used to conduct statistical analyses on the measured data. The data in the figure shows means of four plants ± standard deviation. Tukey Multiple Comparisons test was adopted to compare the differences between treatments. Different lowercase letters for the same parameter indicate significant differences among different treatments at $p < 0.05$ levels.

## 3. Results

### 3.1. Effects of N Metabolism Indicators

$NO_2$ fumigation affected nitrogen metabolism in mulberry. The nitrate nitrogen content of mulberry leaves fumigated with $NO_2$ increased by 56.10% ($p < 0.05$) at 4 h after the start of the treatment, and when the fumigation time lasted for 8 h, it was nearly doubled ($p < 0.05$) compared to the control. The amino acid content in mulberry was not significantly increased ($p > 0.05$) when $NO_2$ was fumigated for 4 hours; however, it increased by 35.17% ($p < 0.05$) after 8 hours of fumigation.

Nitrate reductase and nitrite reductase activities increased significantly ($p < 0.05$) by 4 h after the start of the treatment, and when the fumigation time lasted for 8 h, the activities enhanced more (Figure 2a–d).

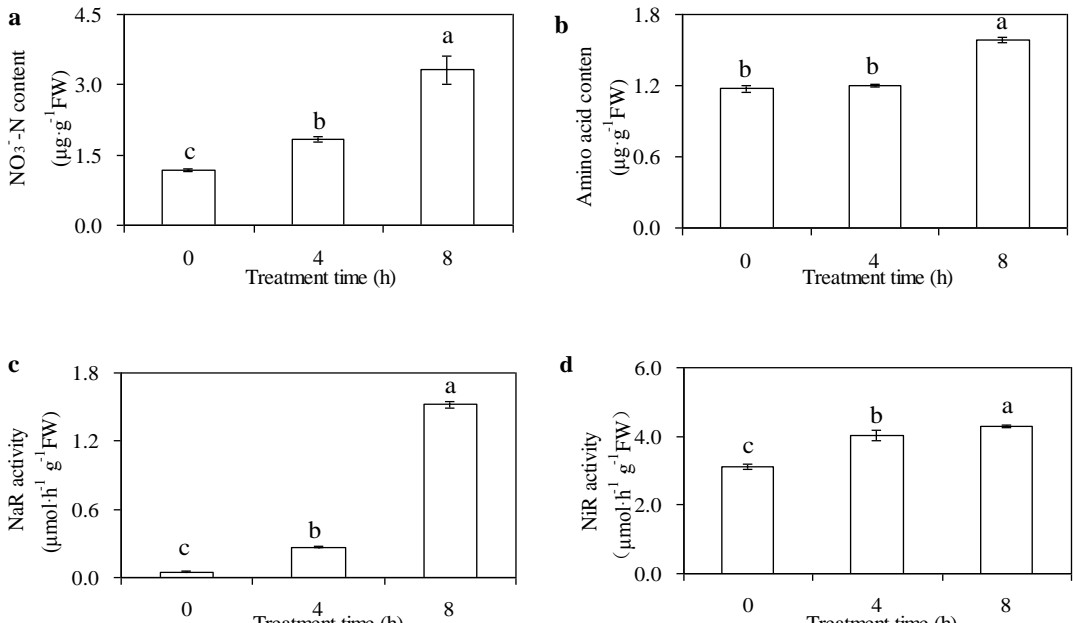

**Figure 2.** Effects of N Metabolism Indicators. (**a**) $NO_3{}^-$-N content, (**b**) amino acid content, (**c**) nitrate reductase activity, and (**d**) nitrite reductase activity in leaves of mulberry seedlings exposure to 4 $\mu L \cdot L^{-1}$ $NO_2$ for 0 h, 4 h (4 $h \cdot d^{-1}$, for one day), and 8 h (4 $h \cdot d^{-1}$, for 2 days). Date represent means of four plants ± standard deviations. Different lowercase letters for the same parameter indicate significant differences among different treatments at $p < 0.05$ levels.

### 3.2. Effects of Gas Exchange Parameters

$NO_2$ fumigation changed the $P_n$-Ci curve of mulberry leaves, and $P_n$ showed an upward trend with the increase of $CO_2$ concentration. When the $CO_2$ concentration was lower than 400 $\mu mol \cdot mol^{-1}$, $P_n$ increases approximately linearly with the increase of $CO_2$ concentration. Subsequently, as the $CO_2$ concentration continued to increase, the rate of $P_n$ increase slowed. When the $CO_2$ concentration reached 1200 $\mu mol \cdot mol^{-1}$, the $P_n$-Ci gradually flattened (Figure 3a). Under the same $CO_2$ concentration, the $P_n$ of mulberry leaves with $NO_2$ fumigation was significantly higher than that of the control, indicating that $NO_2$ fumigation improved the carbon assimilation capability of mulberry leaves.

Stomatal conductance ($G_s$), the maximum carboxylation rate ($V_{cmax}$), and dark respiration rate ($R_d$) of the mulberry leaves increased significantly ($p < 0.05$) after 4 h and 8 h of fumigation compared with the control (Figure 3b–d).

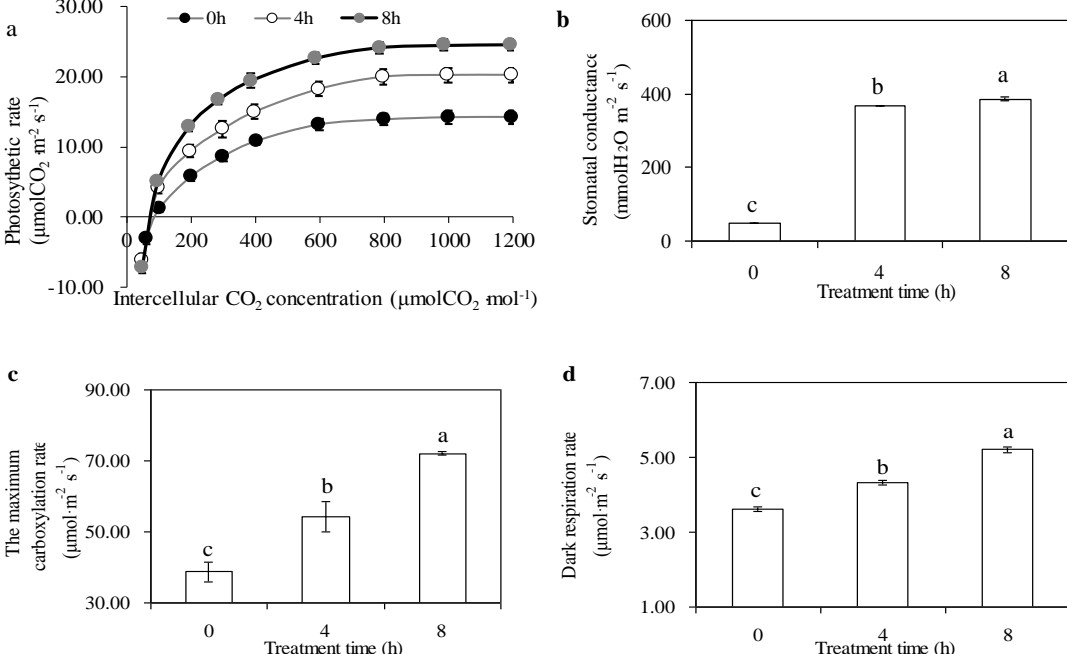

**Figure 3.** Effects of Gas Exchange Parameters. (**a**) $P_n$-Ci curve, (**b**) conductance of $H_2O$, (**c**) the maximum carboxylation rate, and (**d**) dark respiration rate in leaves of mulberry seedlings exposure to 4 $\mu L \cdot L^{-1}$ $NO_2$ for 0 h, 4 h (4 $h \cdot d^{-1}$, for one day), and 8 h (4 $h \cdot d^{-1}$, for 2 days). Date represent means of four plants $\pm$ standard deviations. Different lowercase letters for the same parameter indicate significant differences among different treatments at $p < 0.05$ levels.

## 3.3. Effects of Distribution of Light Absorbed by PSII

Of the light energy absorbed by PSII, the fraction allocated to photochemical conversion ($\Phi_{PSII}$), increased slightly, while the fraction dissipated nonphotochemically in a manner dependent on the trans-thylakoid proton-gradient and the xanthophyll cycle ($\Phi_{NPQ}$) increased after $NO_2$ fumigation. By contrast, the proportion of basic fluorescence quantum yields and heat-dissipated quantum yields ($\Phi_{f,D}$) and heat-dissipation quantum yields of inactive PSII reaction centers ($\Phi_{NF}$) declined after $NO_2$ fumigation. The proportions of $\Phi_{NPQ}$ and $\Phi_{PSII}$ increased by 2.49% ($p < 0.05$), 36.23% ($p < 0.05$) at 4h and increased by 5.67% ($p < 0.05$), 41.07% ($p < 0.05$) at 8h. By contrast, the proportions of $\Phi_{f,D}$ and $\Phi_{NF}$ decreased by 5.01% ($p < 0.05$), 60.41% ($p < 0.05$) at 4h and decreased by 10.34% ($p < 0.05$) and 75.74% ($p < 0.05$) at 8 h (Figure 4). The quantum yield for heat dissipation in mulberry leaves fumigated with 4 $\mu L \cdot L^{-1}$ $NO_2$ was reduced to increase the photosynthetic energy used for photochemical reaction. As shown in Figure 7, the partial energy from total energy absorbed by mulberry leaves fumigated by $NO_2$ was used for $NO_2$ metabolism; whether this affects the activity of PSII and PSI is discussed in the following sections.

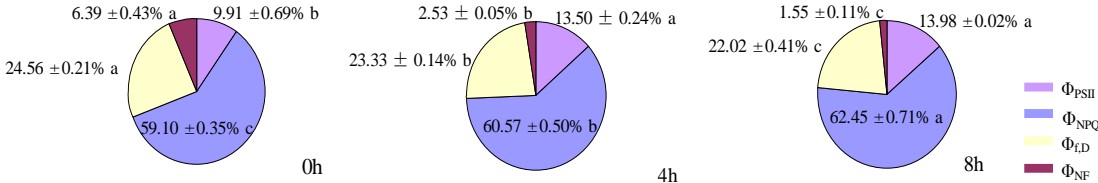

**Figure 4.** Light energy used for quantum yield of PSII photochemistry ($\Phi_{PSII}$), xanthophyll-mediated thermal dissipation ($\Phi_{NPQ}$), basic fluorescence quantum yield and heat dissipation quantum yield ($\Phi_{f,D}$), and quantum yield used in thermal dissipation in non-functional PSII ($\Phi_{NF}$) in leaves of mulberry seedlings exposure to 4 $\mu L \cdot L^{-1}$ $NO_2$ for 0 h, 4 h (4 $h \cdot d^{-1}$, for one day), and 8 h (4 $h \cdot d^{-1}$, for 2 days). Date represent means of four plants $\pm$ standard deviations. Different lowercase letters for the same parameter indicate significant differences among different treatments at $p < 0.05$ levels.

### 3.4. Effects of Chlorophyll A Fluorescence Transient

$NO_2$ fumigation significantly changed the shape of OJIP curves. The fluorescence intensity at points O and J ($F_o$ and $F_J$, respectively) decreased, and the fluorescence intensity at points I and P ($F_I$ and $F_p$, respectively) increased significantly. The OJIP curves of the leaves were normalized to the span O-P. The variable fluorescence at point J on the OJIP curve of leaves was significantly increased by $NO_2$. The difference between the normalized O-P curve and the control was assessed, and the difference between $\Delta Vt$ and the control at point J was most significant at 2 ms (Figure 5a–d).

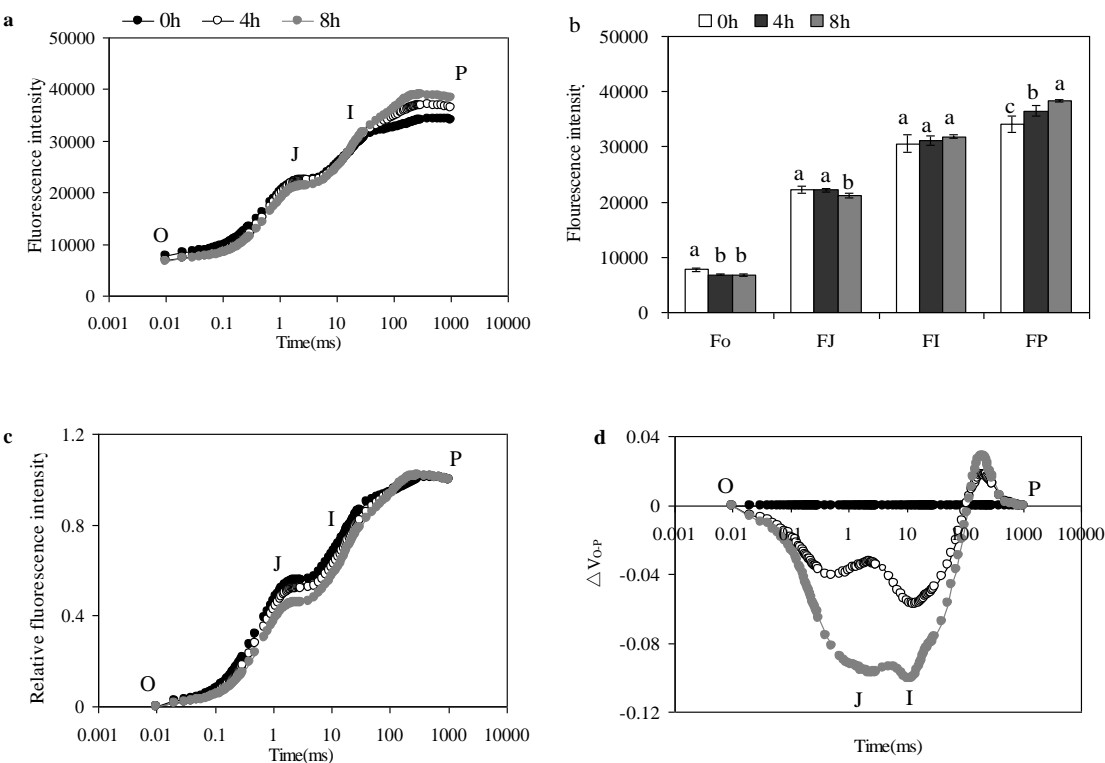

**Figure 5.** Effects of Chlorophyll A Fluorescence Transient. (**a**) OJIP transmission, (**b**) fluorescence intensity at points O, J, I, and P, (**c**) standardized OJIP transmission, (**d**) standardized OJIP transmission and CK in leaves of mulberry seedlings exposure to 4 $\mu L \cdot L^{-1}$ $NO_2$ for 0 h, 4 h (4 $h \cdot d^{-1}$, for one day), and 8 h (4 $h \cdot d^{-1}$, for 2 days). Date represent means of four plants $\pm$ standard deviations. Different lowercase letters for the same parameter indicate significant differences among different treatments at $p < 0.05$ levels.

### 3.5. Effects of PSII activity

NO$_2$ fumigation had a significant effect on PSII potential photochemical activity ($F_v/F_o$), PSII maximum photochemical efficiency ($F_v/F_m$), photochemical quenching (qP), electron transport rate (ETR), and number of active reaction centers per unit area ($RC/CS_m$), which is shown in Table 1. $F_v/F_o$, $F_v/F_m$, qP, ETR, and $RC/CS_m$ increased by 33.44% ($p < 0.05$), 1.22% ($p < 0.05$), 30% ($p < 0.05$), 33.64% ($p < 0.05$), and 27.07% ($p < 0.05$) at 4 h, respectively, and increased by 39.38% ($p < 0.05$), 2.44% ($p < 0.05$), 35% ($p < 0.05$), 37.67% ($p < 0.05$), and 33.37% ($p < 0.05$) at 8 h, respectively.

**Table 1.** PSII potential photochemical activities ($F_v/F_o$), PSII photochemical efficiency ($F_v/F_m$), photochemical quenching (qP), electron transport rate (ETR), and the number of reactive centers per unit area ($RC/CS_m$) in leaves of mulberry seedlings exposure to 4 $\mu$L·L$^{-1}$ NO$_2$ for 0 h, 4 h (4 h·d$^{-1}$, for one day), and 8 h (4 h·d$^{-1}$, for 2 days). Date represent means of four plants ± standard deviations. Different lowercase letters for the same parameter indicate significant differences among different treatments at $p < 0.05$ levels.

|  | 0 h | 4 h | 8 h |
|---|---|---|---|
| $F_v/F_o$ | 3.20 ± 0.50b | 4.27 ± 0.12a | 4.46 ± 0.32a |
| $F_v/F_m$ | 0.82 ± 0.00c | 0.83 ± 0.00b | 0.84 ± 0.01a |
| qP | 0.20 ± 0.05b | 0.26 ± 0.01a | 0.27 ± 0.00a |
| ETR | 43.99 ± 3.32b | 58.79 ± 0.92a | 60.56 ± 0.17a |
| $RC/CS_m$ | 20501.38 ± 377.35c | 26050.96 ± 1194.94b | 27341.97 ± 206.688a |

### 3.6. Effects of PSI Activity

The relative drop in 820 nm light signals during far-red light illumination reflects the activity of PSI $\Delta I/I$o. NO$_2$ fumigation increased the drop in the 820 nm optical signal, and the difference increased significantly at 8 h, indicating that NO$_2$ fumigation increased the PSI activity of mulberry leaves (Figure 6a). PSI activity was not significantly raised ($p > 0.05$) for 4 h fumigation, but when fumigated for 8 h, the PSI activity was improved by 20.22% ($p < 0.05$) compared with the control (Figure 6b).

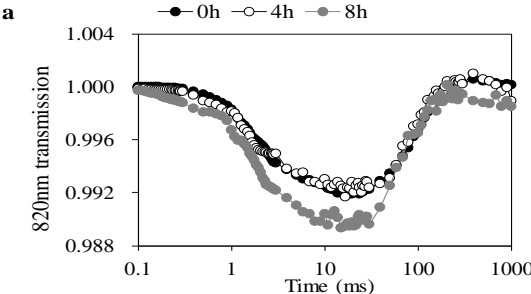 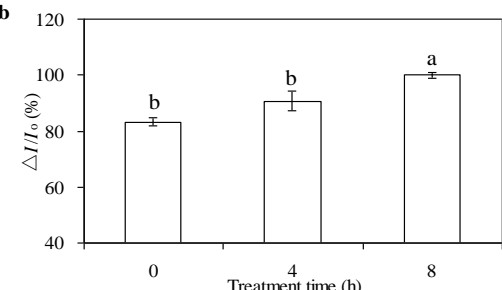

**Figure 6.** Effects of PSI activity. (**a**) Changes in light transmission at 820 nm, (**b**) Relative values of $\Delta I/I_o$ in leaves of mulberry seedlings exposure to 4 $\mu$L·L$^{-1}$ NO$_2$ for 0 h, 4 h (4 h·d$^{-1}$, for one day), and 8 h (4 h·d$^{-1}$, for 2 days), where the $\Delta I/I_o$ values at 8 h were taken as 100%. Date represent means of four plants ± standard deviations. Different lowercase letters for the same parameter indicate significant differences among different treatments at $p < 0.05$ levels.

## 4. Discussion

The present study revealed that the absorption of NO$_2$ by mulberry leaves is not only involved in nitrogen metabolism in vivo, but also increases the leaf content of nitrate nitrogen and amino acids, enhances the activity of nitrate reductase (NR) and nitrite reductase (NiR), and improves the nitrogen metabolism capacity (Figure 2). Zeevaart et al. [39] fumigated pea seedlings with ammonium nitrogen as the only nitrogen source condition and found that NO$_2$ induces NR activity, in perhaps the earliest study on NO$_2$ and nitrogen metabolism. Subsequent studies have since found that atmospheric

$NO_2$ has a significant effect on NR. Lower concentrations of $NO_2$ increase NR activity in barley (*Hordeum vulgre*) [40], Scots pine (*Pinus sylvestris*) [41], and red spruce (Picea rubens) [42]. Similarly, when Hisamatsu et al. [43] fumigated squash (*Cucurbita maxima*) seedlings with $NO_2$, NR activity in cotyledon was significantly reduced. In the present study, NR activity had exhibited a small increase at 4 h but increased sharply at 8 h. Enhanced NR activity promotes the improvement of nitrogen metabolism. It not only increases the consumption of excess light energy, but also provides necessary enzymes for carbon metabolism and accelerates carbon metabolism.

The 4 $\mu L\cdot L^{-1}$ $NO_2$ fumigation treatment significantly increased the net photosynthetic rate of mulberry leaves (Figure 3a), accelerating photosynthetic electron transport and enhancing phosphorylation activity in leaves. Additionally, the increase of stomatal conductance promoted the degree of $CO_2$ acquisition and transportation (Figure 3b), thereby accelerating the production and accumulation of organic matter in plants [44]. At the same time, the maximum carboxylation rate, which is an important parameter for characterizing photosynthetic capacity of plants, also increased (Figure 3b), indicating that $NO_2$ fumigation increased Rubisco activity of mulberry leaves, thereby enhancing the fixation ability of $CO_2$ [45]. Moreover, the dark respiration rate was significantly increased (Figure 3d), indicating that its fumigated leaves consumed excess light energy through respiration to protect the photosynthetic system and increase the photosynthetic rate.

Accordingly, the quantitative study of the final destination of light energy absorbed by plant leaves is an important part of research on photosynthesis. Earlier studies on light energy are more directly expressed by actual photochemical efficiency ($\Phi_{PSII}$) and non-photochemical quenching (NPQ) [46,47]. However, in higher plants, NPQ is only determined by the establishment of proton gradients on both sides of the thylakoid membrane and the xanthophyll cycle [48]. Thus, NPQ does not represent all non-photochemical quenching processes. In addition, plant leaves absorb light energy via mechanisms other than photochemical reactions, including physiological processes such as photorespiration, the water–water cycle, and xanthophyll cycle, which can be areas in which light energy is distributed as the final destination [49–51]. The theory developed by Hendrickson et al. [50] clarifies these processes. In this framework, excitation energy can be divided into the light energy absorbed by the PSII reaction centers and used for the quantum yield of photochemical reaction ($\Phi_{PSII}$), the quantum production dependent on the trans-thylakoid proton-gradient for xanthophyll cycle quantum yield ($\Phi_{NPQ}$), fluorescence quantum yield and heat dissipation ($\Phi_{f,D}$), and heat dissipation quantum yield in inactive PSII reaction centers ($\Phi_{NF}$) [51]. In our experiment, mulberry leaves fumigated with $NO_2$ showed proportional increases in $\Phi_{PSII}$ and $\Phi_{NPQ}$, indicating that $NO_2$ fumigation promoted the establishment of light-induced proton absorption in the chloroplast $H^+$ concentration gradients on both sides of the thylakoid. In other words, the presence of $\Delta pH$ on both sides of the thylakoid membrane increased the power of ATP synthesis in chloroplasts and promoted the photoprotective mechanism based on the xanthophyll cycle. Additionally, $\Phi_{f,D}$ and $\Phi_{NF}$ proportionally decreased, indicating that the proportion of heat dissipation and inactivation reaction centers decreased and that the absorbed light energy was mostly used for photosynthetic carbon assimilation, thereby improving the photochemical efficiency of the mulberry leaf (Figure 4).

In this experiment, the changes in PSII of mulberry leaves were analyzed using the JIP test. $NO_2$ changed the structure and function of PSII and the photosynthetic primary reaction process in mulberry leaves. However, because the OJIP curve is greatly influenced by the environment, its relative fluorescence intensity is affected by various environmental factors. Therefore, the OJIP curve was often normalized to the span O-P. The relative variable fluorescence of $V_J$ and $V_I$ at points J (2 ms) and I point (30 ms) for $NO_2$-fumigated leaves was significantly decreased (Figure 5), indicating that the PSII reaction center receptor-side electronic primary quinone receptors ($Q_A$) to the secondary quinone receptor ($Q_B$) transmission capacity and plastoquinone (PQ) accept electronic ability were enhanced (Figure 7). The potential photochemical activity of PSII ($F_v/F_o$), PSII maximum photochemical efficiency ($F_v/F_m$), photochemical quenching (qP), electron transfer rate (ETR), and the number of active reaction centers per unit area ($RC/CS_m$) were significantly higher

under fumigation than in the control treatment (Table 1), indicating that $NO_2$ fumigation enhances the activity of PSII reaction centers and the degree of openness of reaction centers, which promotes the electron transport of leaves, photosynthetic primary reaction process, and the rate of light photons received by PSII reaction centers. The proportion of light energy used in photochemical reactions increased, thus accelerating the synthesis of NADPH and ATP, as well as the conversion efficiency of light energy and carbon assimilation processes, which increased the photochemical efficiency. However, Hu et al. [31] quantified the photosynthetic responses of hybrid poplar cuttings to 4 $\mu L \cdot L^{-1}$ $NO_2$. It was found that significant declines in $F_v/F_m$, indicating inhibition of and even damage to photosynthetic apparatus. The study of the maximum redox capacity ($\Delta I/Io$) of PSI was also increased (Figure 6) as well as the ability to receive electrons; the ability of PSII to supply electrons in the photosynthetic apparatus was matched by the ability of PSI to receive electrons. Thus, not only can electron transfer be promoted efficiently, but electrons can also be transferred to ferredoxin, which distributes electrons to the nitrogen metabolism, photorespiration, and $NO_2$ metabolic pathways (Figure 7).

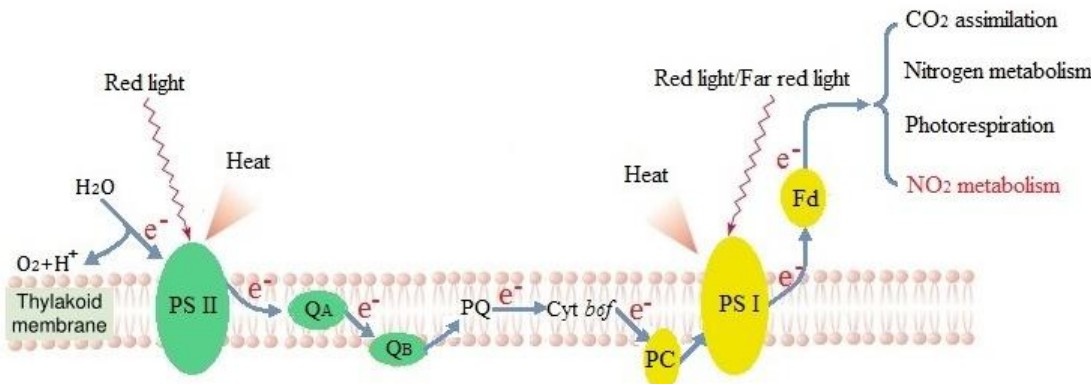

**Figure 7.** Photosynthetic electron flow distribution. The electrons produced by water splitting in photosynthesis pass through photosystem II (PSII), primary quinone receptor ($Q_A$), secondary quinone receptor ($Q_B$), plastoquinone (PQ), cytochrome $b_6f$ (Cyt $b_6f$), plastid blue pigment (PC), and photosystem I (PSI) to ferredoxin (Fd) were assigned to four pathways: $CO_2$ assimilation, nitrogen metabolism, photorespiration, and $NO_2$ metabolic pathways (new pathway).

## 5. Conclusions

The metabolism of atmospheric $NO_2$ utilized and consumed light energy and photosynthetic electrons absorbed by leaves of mulberry fumigated with $NO_2$. However, this part of the photosynthetic energy consumption did not reduce the photosynthetic capacity of the mulberry leaves, but instead increased the photosynthetic efficiency of the plant leaves. This is because the mulberry leaves absorbed $NO_2$ and conducted nitrogen metabolism and respiration, which consumed excess light energy, and thus protected the photosynthetic apparatus. However, the light energy absorbed by the PSII reaction center in the form of heat dissipation in mulberry plants fumigated with 4 $\mu L \cdot L^{-1}$ $NO_2$ was also reduced, such that the absorbed light energy was more effectively used in photosynthetic carbon assimilation. Therefore, in the case where the concentration of the atmospheric pollutant $NO_2$ is lower than the concentration that can damage the mulberry, $NO_2$ can be absorbed by the mulberry to reduce the haze in the air.

**Author Contributions:** Y.W. and G.S. conceived and designed the study. Y.W., W.J., Y.C., and D.H. performed the experiments. Y.W., J.W., and M.Z. contributed to the sample measurement and data analysis. Y.W. and G.S. wrote the paper.

**Funding:** This research was funded by the National Natural Science Foundation of China, grant number 31870373, and the Natural Science Foundation of Heilongjiang Province, grant number ZD201105 and the Applicant and Developmental Project for Agriculture of Heilongjiang Province, grant number GZ13B004.

**Acknowledgments:** The authors want to appreciate Wah Soon Chow from the Australian National University for revising the manuscript and our colleague Yanbo Hu for his advice and great comments to improve the paper.

**Conflicts of Interest:** The authors declare no conflict of interest.

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
