# Peer review of "Atmospheric Nitrogen Dioxide Improves Photosynthesis in Mulberry Leaves via Effective Utilization of Excess Absorbed Light Energy"

_forests, doi:10.3390/f10040312_

Round 1

Reviewer 1 Report

attached file

Author Response

Point 1: This manuscript provides new interesting information that the mulberry leaves absorbed NO2 and conducted nitrogen metabolism and respiration, which consumed excess light energy and thus protected the photosynthetic apparatus. The manuscript is well written and the methods used are appropriate.

Response 1: We great thanked the reviewer’s valuable comments.

Point 2: Section 2.9., Lines 227, 239: The expression quantum yield of fluorescence quantum yield and heat dissipation (Φf,D)needs to be corrected with " quantum yield of fluorescence and thermal/heat energy dissipation (Φf,D)” or similar (see Hendrickson et al., 2005).

Response 2: Thanks for the comment. Authors have modified in the re-submission and highlighted in yellow. Please see Section 2.9., Lines 233, 245.

Point 3: Line 253: “increased 56.08% (P < 0.01) by 4 h after the start of …” should be “increased by 56.08% (P < 0.01) at 4 h after the start of …

Response 3: Thanks for the comment. Authors have modified in the re-submission and highlighted in yellow. Please see Lines 258.

Point 4: Figure 2d: The inscription on the Y axis NiR activity / μmol·h-1·g-1FW should be given with brackets: NiR activity (μmol·h-1·g-1FW)”  as in Figures 2a-c.

Response 4: We are very sorry for our negligence and have corrected the Figure.

Point 5: Figures 3a-d: Units should be given in brackets, e.g. Dark respiration rate (μmol·m-2·s-1)”.

Response 5: We have corrected Figures according to the Reviewer’s comment.

Point 6: Figure 3b:  “Conductance to H2O/ mol H2O·m-2·s-1” should be “Stomatal conductance (mol H2O·m-2·s-1)”

Response 6: Thanks for the valuable comments. Authors have modified the Figure.

Point 7: Lines 294-295: The sentence “The proportions of ΦNPQ and ΦPSII increased by 36.39% (P < 0.01), 2.45% (P < 0.01) at 4h and increased by 40.93% (P < 0.01), 5.62% (P < 0.01) at 8h“ (Figure 4).  Authors should check again the percentages for ΦNPQ (36.39% and 40.93%) or to give more detailed explanations in the test, because the results shown on Figure 4 for ΦNPQ  at 0h are 59.10 %, at 4h are 60.57 %, at 8h are 62.45 %.

Response 7: We are very sorry for our incorrect writing. We have rechecked all the data and modified it. 

Point 8: Table 1. Authors should also check the percentages for Fv/Fm (e.g.  6.71% (P < 0.01) and 7.40% (P < 0.01), which must correspond to the results shown in Table 1 for this parameter (0.82, 0.83 and 0.84), which are almost unchanged.

Response 8: Thanks for the comment. We are very sorry for our negligence. We have rechecked all the data and modified it.

Reviewer 2 Report

Your research was novel, and results had a value to publish in Forests. However, I could not confirm the period of fumigation of nitrogen dioxide. In theory, your experiment could conduct only one day of fumigation. In my opinion, the period of fumigation needs at least one month. If your establishment of fumigation was insufficient, your results was no value to publish in Forests. Woody species grow for a long period, and reaction of nitrogen dioxide should examine for a long period. In the present circumstances, I decided that your submission needed major revision.

Also, I would like to indicate some points for improvement as following;

1.     You did not introduce the actual situation of damage of plant by NO2 in China. You should explain the importance of your experiment from the problem of emission of NO2.

2.     You did not introduce the effects of mulberry to the atmospheric pollutants. There might be no case the effects of NO2 for mulberry. However, there is a possibility that mulberry was examined effects of other pollutants. This information was important whether mulberry was sensible to pollutants or not.

3.     You did not mention how many seedlings did you used for three treatments.

4.     You did not explain the reason why you fumigate NO2 for four and eight hours.

5.     On the statistical analysis, your manuscript had problems. According to the results, your statistical analysis was considered as a multiple comparison. However, you could not use ANOVA for multiple comparison. You should change into other statistical analysis (e.g. Tukey test). Also, I could not understand the reason why you used capital and small letters. You should use only small letters.

6.     On your discussion, there was many descriptions to explain mechanisms of photosynthetic characteristics (e.g. P. 10, L. 351-355; P. 11, L. 401-403). These descriptions were superfluous for your discussion. At first, you should write your discussion according to your results for each paragraph. Also, you should quote figures or tables in your discussion.

7.     On the chapter of chlorophyll fluorescence (P. 11, L. 401-424), there was no quotation of other experiment. You should discuss by the comparisons of other research.

8.     Your conclusion was summary of results. I could not understand what was concluded from your results. Also, you should predict the effects of mulberry for emission of NO2 in China.

Other minor comments for revision

P. 1, L. 39-41: You should quote literatures.

P. 3, L. 97: There was no information of the size of pot

Fig. 2 b, d: The division of interval of vertical axis should change (e.g. 50, 100, 150, 200; 2, 4, 6)

Fig. 2 c, d: On the title of vertical axis, you should uniform the size of alphabet (g)

Fig. 3, all of figures: You should uniform the size of alphabet on the title of vertical axis.

Fig. 3 a: The interval of horizontal axis was not regular. You should change into actual values. On the Pn-Ci curve, you have to draw according to the equation (e.g. Farquhar and Sharkey, 1982, Annual Review Plant Physiology)

Fig. 3 c: On the interval of vertical axis, you should change into regular interval.

Fig. 7: You should explain this figure in previous chapter because this mechanism was important.

Author Response

Point 1: Your research was novel, and results had a value to publish in Forests. However, I could not confirm the period of fumigation of nitrogen dioxide. In theory, your experiment could conduct only one day of fumigation. In my opinion, the period of fumigation needs at least one month. If your establishment of fumigation was insufficient, your results was no value to publish in Forests. Woody species grow for a long period, and reaction of nitrogen dioxide should examine for a long period. In the present circumstances, I decided that your submission needed major revision.

Response 1: We great thanked the reviewer’s valuable comments. In this experiment, the fumigation time was 4h and 8h, which was because the acute response of plants was the most significant at this time, and the results obtained by exploring the response mechanism of the photosynthetic characteristics of mulberry leaves to atmospheric NO2 were the most significant. However, long-term fumigation, such as several weeks or months, belongs to the adaptive response of plants. Acute response and adaptive response also have research value and significance. In this paper, we studied the acute response of photosynthetic characteristics in mulberry leaves to atmospheric NO2, and its adaptive response to atmospheric NO2 is also being studied.

Point 2: You did not introduce the actual situation of damage of plant by NO2 in China. You should explain the importance of your experiment from the problem of emission of NO2.

Response 2: Thanks for the comment. Authors have added these information in the re-submission. Please see P. 2, L. 63-65.

Point 3: You did not introduce the effects of mulberry to the atmospheric pollutants. There might be no case the effects of NO2 for mulberry. However, there is a possibility that mulberry was examined effects of other pollutants. This information was important whether mulberry was sensible to pollutants or not.

Response 3: Thanks for the valuable comment. We have added these information in the re-submission according to the Reviewer’s comments. Please see P. 2, L. 78-80.

Point 4: You did not mention how many seedlings did you used for three treatments.

Response 4: Thanks for the comment. Authors have added this part in the re-submission. Please see P. 3, L. 104-105.

Point 5: You did not explain the reason why you fumigate NO2 for four and eight hours.

Response 5: Thanks for the valuable comment. In this experiment, we wanted to explore the response mechanism of the photosynthetic characteristics of mulberry leaves to atmospheric NO2 through the acute response of mulberry trees to atmospheric NO2, so we chose short time fumigation. In the pre-experiment, we fumigated for 2 h, 4 h, 6 h, 8 h and 10 h, respectively, and finally chose the 4h and 8h with the most significant results.

Point 6: On the statistical analysis, your manuscript had problems. According to the results, your statistical analysis was considered as a multiple comparison. However, you could not use ANOVA for multiple comparison. You should change into other statistical analysis (e.g. Tukey test). Also, I could not understand the reason why you used capital and small letters. You should use only small letters.

Response 6: Thanks for the comment. Author has changed the statistical analysis of ANOVA to Tukey test., and only lowercase letters have been used in the re-submission. Please see P. 6, L. 251-252, Figure 2-6, Table 1.

Point 7: On your discussion, there was many descriptions to explain mechanisms of photosynthetic characteristics (e.g. P. 10, L. 351-355; P. 11, L. 401-403). These descriptions were superfluous for your discussion. At first, you should write your discussion according to your results for each paragraph. Also, you should quote figures or tables in your discussion.

Response 7: We have re-written this part according to the Reviewer’s suggestion. Please see P. 10, L. 360-P. 11, L. 429.

Point 8: On the chapter of chlorophyll fluorescence (P. 11, L. 401-424), there was no quotation of other experiment. You should discuss by the comparisons of other research.

Response 8: We great thanked the reviewer’s valuable comment and added the research results of Hu and discussed with our experiment on the chapter of chlorophyll fluorescence. Please see P. 11, L. 422-424. Hu, Y.; Bellaloui, N.; Sun, G.; Tigabu, M.; Wang, J. Exogenous sodium sulfide improves morphological and physiological responses of a hybrid populus, species to nitrogen dioxide. J. Plant Physiol. 2014, 171, 868-875.

Point 9: Your conclusion was summary of results. I could not understand what was concluded from your results. Also, you should predict the effects of mulberry for emission of NO2 in China.

Response 9: Thanks for the valuable comment. Authors had revised conclusion according to the reviewer’s comments. Please see P. 12, L. 447-449.

Point 10: P. 1, L. 39-41: You should quote literatures.

Response 10: Thanks for the comment. Authors had added literature in the re-submission.

Fang, H.; Mo, J.M. Reactive nitrogen increasing: A threat to our environment. Ecology and Environment Sciences. 2006, 15, 164-168.

Point 11: P. 3, L. 97: There was no information of the size of pot

Response 11: Thanks for the reviewer’s comment. The size of pot had been described in P. 3, L.96-97 in the original.

Point 12: Fig. 2 b, d: The division of interval of vertical axis should change (e.g. 50, 100, 150, 200; 2, 4, 6)

Response 12: Thanks for the comments. Authors have modified the Figure according to the Reviewer’s comment.

Point 13: Fig. 2 c, d: On the title of vertical axis, you should uniform the size of alphabet (g)

Response 13: We are very sorry for our negligence and have modified the Figure.

Point 14: Fig. 3, all of figures: You should uniform the size of alphabet on the title of vertical axis.

Response 14: Thanks for the valuable comments. Authors have modified the Figure.

Point 15: Fig. 3 a: The interval of horizontal axis was not regular. You should change into actual values. On the Pn-Ci curve, you have to draw according to the equation (e.g. Farquhar and Sharkey, 1982, Annual Review Plant Physiology)

Response 15: Thanks for the valuable comments. Authors have modified the Figure and re-draw the Pn-Ci curve according to the equation of Farquhar and Sharkey

Point 16: Fig. 3 c: On the interval of vertical axis, you should change into regular interval.

Response 16: We are very sorry for our negligence and have modified the Figure.

Point 17: Fig. 7: You should explain this figure in previous chapter because this mechanism was important.

Response 17: We great thanked the reviewer’s valuable comment and added to the interpretation of Figure 7 in the chapter of Effects of Distribution of Light Absorbed by PSII. Please see P. 8, L. 303-307.

Round 2

Reviewer 2 Report

Your manuscript was improved according to my suggestion. However, I don’t agree the experiment of one-day fumigation. Meanwhile, I understand the novelty of originality of your data. Therefore, I would like to depend on the decision of acceptance to the Editor. Also, you have to improve the text of some figures.

Fig. 2 d: On the title of vertical axis, you should uniform the size of alphabet (g)

Fig. 3 a: You should move the horizontal axis to the bottom of this figure. Also, you should change the division of interval of vertical axis (e.g. -10, 0, 10, 20, 30)

Fig. 3 (all): On the title of vertical axis, you should uniform the size of alphabet (e.g. H2Om-2s-1)

Fig. 7: I would like to confirm your opinion on the position of this figure. Do you want to fix at this position?

Author Response

Point 1:Your manuscript was improved according to my suggestion. However, I don’t agree the experiment of one-day fumigation. Meanwhile, I understand the novelty of originality of your data. Therefore, I would like to depend on the decision of acceptance to the Editor. Also, you have to improve the text of some figures.

 Response 1: We great thanked the reviewer’s valuable comment.

Point 2: Fig. 2 d: On the title of vertical axis, you should uniform the size of alphabet (g)

 Response 2: Thanks for the valuable comments. Authors have modified the Figure.

Point 3: Fig. 3 a: You should move the horizontal axis to the bottom of this figure. Also, you should change the division of interval of vertical axis (e.g. -10, 0, 10, 20, 30)

 Response 3: Thanks for the comments. Authors have modified the Figure according to the Reviewer’s comment.

Point 4: Fig. 3 (all): On the title of vertical axis, you should uniform the size of alphabet (e.g. H2Om-2s-1)

 Response 4: Thanks for the reviewer’s comment. Authors have modified the Figure.

Point 5: Fig. 7: I would like to confirm your opinion on the position of this figure. Do you want to fix at this position?

 Response 5: Thanks for the reviewer’s valuable comments. The photosynthetic energy distribution was summarized in discussion section when NO2 was absorbed by the mulberry leaves, and authors believe that it is appropriate to place Fig.7 in the end of discussion section.
